# The Relationship between the Number of Daily Health-Related Behavioral Risk Factors and Sleep Health of the Elderly in China

**DOI:** 10.3390/ijerph16244905

**Published:** 2019-12-04

**Authors:** Xiaojun Liu, Jingshu Chen, Jiayi Zhou, Jianjian Liu, Chanida Lertpitakpong, Anran Tan, Shaotang Wu, Zongfu Mao

**Affiliations:** 1School of Health Sciences, Wuhan University, Wuhan 430071, China; xiaojunliu@whu.edu.cn (X.L.); 2017302180144@whu.edu.cn (J.Z.); jianjianliu@whu.edu.cn (J.L.); chloetar@whu.edu.cn (A.T.); 2Global Health Institute, Wuhan University, Wuhan 430072, China; 3School of Public Health and Management, Hubei University of Medicine, Shiyan 442000, China; chjingshu@163.com; 4Department of Public Health Administration, Faculty of Public Health, Mahidol University, Bangkok 10400 Thailand

**Keywords:** unhealthy daily behaviors, sleep disorder, sleep quality, association, Chinese elderly

## Abstract

This study examined the cross-sectional association among a number of daily health-related behavioral risk factors and sleep among Chinese elderly. A sample of 4993 adults, aged 60 years and older, from the China’s Health-Related Quality of Life Survey for Older Adults 2018 was included in this study. Five daily health-related behaviors, which included smoking, drinking, unhealthy eating habits, insufficient leisure activities, and physical inactivity were measured. Sleep disturbances and sleep quality were used to represent the respondents’ sleep status. Multiple logistic regression models and multiple linear regression models were established. The odds ratios (ORs) of sleep disturbances for those with one to five health-related risk behaviors were 1.41 (95% CI = 1.11 to 1.78), 2.09 (95% CI = 1.66 to 2.63), 2.54 (95% CI = 1.99 to 3.25), 2.12 (95% CI = 1.60 to 2.80), and 2.49 (95% CI = 1.70 to 3.65), respectively. Individuals with one health-related risk behavior (B = 0.14, 95% CI = −0.23 to −0.06), two health-related risk behaviors (B = 0.21, 95% CI = −0.30 to −0.13), three health-related risk behaviors (B = 0.46, 95% CI = −0.55 to −0.37), four health-related risk behaviors (B = 0.50, 95% CI = −0.62 to −0.39), and five health-related risk behaviors (B = 0.83, 95% CI = −1.00 to −0.66) showed lower scores of self-perceived sleep quality. Having multiple health-risk behaviors was positively correlated with a higher risk of sleep disturbances among Chinese elderly. Moreover, elderly individuals with multiple health-related risk behaviors were significantly associated with poorer sleep quality.

## 1. Introduction

Sleep accounts for one-third of an individual’s whole life span, which is essential to an individual’s physical and mental health [1,2,3]. Sleep is a kind of spontaneous physiological and psychological state that occurs periodically in human beings and is mainly manifested as a quiet state of the body and a temporary interruption of consciousness [3]. Sleep is a vital component of life, which helps the body better adapt to the alternation of day and night. Good sleep quality not only eliminates tiredness and improves memory, but also regulates body functions, and therefore maintains health [4], however, sleep problems are extremely common, gradually eliciting a global health concern [1,2,5,6]. Sleep problems, such as sleep disorders can lead to memory decline, lack of concentration, low spirits, and even mental illnesses, including depression and anxiety [4,7,8,9,10,11]. Moreover, adverse effects on the body’s immune and endocrine systems have been found to be due to sleep disorders [12]. According to related studies, sleep disturbances are closely related to cardiovascular diseases, [13] malignant tumors, [14] and other chronic noncommunicable diseases [15].

Existing studies provide several socio-ecological explanations of sleep, which generally believe that sleep can be influenced by psychological and physiological factors, environment, financial status, and other social factors [16,17,18,19,20]. Anxiety could directly lead to poor sleep quality, and there is a strong association between depression symptoms and sleep disturbances [21]. In addition, physiological factors, such as age, gender, and diseases can affect sleep status [17,19,20]. A previous study revealed that the risk of sleep disorders was higher among females, and their sleep qualities were also reported to be worse [22]. Sleep is also largely influenced by diseases. Numerous studies have confirmed the close association between chronic noncommunicable diseases and the prevalence of sleep disorders [13,15]. Meanwhile, it has been shown that medicines used to treat diseases, such as cardiovascular drugs and antipsychotic drugs, have also been related to sleep status [23]. Moreover, lifestyles including tobacco use and alcohol drinking, have been shown to have a great impact on sleep problems and sleep quality [2,16,17,24]. Related studies have shown that there is a clear correlation between the consumption of coffee and sleep problems, and eating problems and weight are also correlated with sleep health [25,26,27]. Because of the co-occurrence effects of behaviors, individuals often have more than one behavior at the same time, which indicates they may have multiple healthy or unhealthy behaviors. Thus, it is insufficient to interpret sleep status by only using the data of one single behavior, however, limited studies have focused on the relationship between sleep and the number of health-related behaviors.

Elderly people are considered to be a vulnerable population. Their risks of sleep disturbances are significantly higher than that of other age groups [19,28]. It has been reported that sleep quality becomes worse with an increase of age, due to a decline in physiological functioning and a change of the endocrine system [3,6,9,10,11]. Moreover, in elderly people, decline in physical function and activities, as well as impairment of self-care ability and activities of daily life aggravate the risk of sleep disorders. A study of sleep structure and longevity of the elderly revealed that total sleep time and sleep efficiency present a significant age-related decrease, [29] which further explained the negative effect that age had on sleep quality. With regards to China, a country with the largest number of elderly people in the world, the aging population has engendered a great health burden, including sleep-related health problems in the elderly.

Nevertheless, there are few studies that focus on sleep-related health issues in mainland China. There is no national data on the prevalence of sleep health, such as insomnia, sleep apnoea, and other sleep disorders, nor have studies specifically targeted at sleep problems of the elderly been conducted. Especially, there is no study reporting an association between sleep and the number of daily health-related behavioral risk factors. Therefore, this study explores the relationship between sleep health and the number of health-related behaviors, and therefore provides a theoretical basis for the implementation of relevant, effective and targeted intervention programs for preventing risky behaviors to improve the quality of life and health status of the elderly.

## 2. Materials and Methods

### 2.1. Study Population

The derivation sample for this study came from China’s Health-Related Quality of Life Survey for Older Adults 2018 (CHRQLS-OA 2018). In China, the Health-Related Quality of Life Survey for Older People 2018 was conducted by the Global Health Institute of Wuhan University during the Spring Festival in 2018, and aimed to collect data on the socio-ecological factors and health status of the elderly in China, including the participants’ individual socio-demographic characteristics, social capital, behaviors and lifestyles, health-related quality of life, mental health and coping strategies, etc. All participants were 60 years old and over. Using a convenience sampling strategy, a general database containing 5442 valid samples was finally established. The survey is a cross-sectional population-based survey conducted both online and offline during the Spring Festival, when the population is most evenly distributed in China. The response rate for the offline survey was 85.26%.

In this study, 4993 individuals, aged 60 years or above, were included in the final analysis. Because this study aimed to examine the cross-sectional association between the number of daily health-related behavioral risk factors and sleep, among the elderly in China, we excluded subjects with no relevant information about daily health-related behavioral risk factors and sleep. There were 449 unqualified subjects, which accounted for 8.25% of the total number of participants.

### 2.2. Description of Measures

#### 2.2.1. General Demographic Characteristics

For this study, the following general demographic characteristics of the participants were included: age, sex, body mass index (BMI), years of education, household registration, marital status, self-rated family financial situation, and self-rated health status. These general demographic characteristics were considered as confounding factors in the examination of the cross-sectional association between the number of daily health-related behavioral risk factors and sleep among Chinese elderly.

#### 2.2.2. Assessment of Health-Related Risk Behaviors

Five daily health-related behavioral risk factors were measured in this study. In China’s, Health-Related Quality of Life Survey for Older Adults 2018, participants were invited to answer questions on their current status of daily health-related behaviors, which included smoking, drinking, eating behaviors, leisure activities, and physical exercise. According to the Healthy China Action Plan (2019–2030) [30] and the purposes of this study, health-related risk behaviors were defined as follows: (1) current smoking, i.e., participants who self-reported smoking at least one cigarette per week were defined as current smokers, while individuals who have never smoked or have quit smoking were current nonsmokers; (2) current drinking, i.e., individuals who self-reported drinking more than one time per week were defined as current drinkers, and those who never drank in the past were current nondrinkers; (3) unhealthy dietary behavior, i.e., individuals who self-reported skipping breakfast or having an unbalanced diet such as an insufficient intake of vegetables and fruit; (4) insufficient leisure activities participation, i.e., individuals who self-reported never participating in playing cards, Mahjong, chess, etc.; and (5) physical inactivity, i.e., individuals who did not meet the standard set by the Chinese Center for Disease Control and Prevention (CDC), i.e., doing exercise more than three times per week and at least 30 minutes per time, were identified as physical inactivity. Finally, we calculated the total number of these five health-related risk behaviors. 

#### 2.2.3. Assessment of Sleeping Status

In China’s Health-Related Quality of Life Survey for Older Adults 2018, sleeping status was determined by asking participants to self-report on whether they have any sleep problems and to self-evaluate the quality of their sleep. The respondents were asked to answer the question, “have you had the following sleep problems in the past 30 days: (1) do not have any sleep problem, (2) have trouble falling asleep, (3) dreaminess, (4) frequently wake up at night, (5) oversleep, and (6) others?” If a participant chooses the answer, “(1) do not have any sleep problem”, then they went to the next question, otherwise they could choose more than one answer. As a semi-closed question, participants were asked to clarify specific sleep problems when choosing the answer “(6) others”. In the final statistical analysis, we combined answers to “2 to 6” to perform a comparison with answer “1”. The participants’ self-perceived sleep quality was measured with one single standardized question “how do you rate the quality of your sleep in the past 30 days?” This question had five classified ordinal variables as follows: very poor, 1; poor, 2; moderate, 3; good, 4; and very good, 5. Finally, we calculated the average score of the participants’ self-rated sleep quality. 

### 2.3. Research Design and Participants

The Statistical Package for the Social Sciences (SPSS) version 23.0 for Windows (SPSS Inc., Chicago, IL, USA) was employed to run all statistical analyses, with a statistical significance level of 0.05. 

Data analysis was performed in four steps. First, general demographic characteristics of the participants were summarized via initial descriptive analysis with frequencies and proportions. Secondly, the chi-square test and t-test were used to compare differences in the distribution of the number of health-related risk behaviors, and self-perceived sleep quality between the two groups with and without sleep disturbances. Third, multiple logistic regression models were used to identify the relationship between the number of daily health-related behavioral risk factors and sleep disturbance. The results were presented as an odds ratio (OR) value with a 95% confidence interval (95% CI). Finally, multiple linear regression analysis was used to examine the association between the number of daily health-related behavioral risk factors and level of self-perceived sleep quality. The unstandardized coefficients (B) with a 95% confidence interval (95% CI) obtained from the model were reported. We established three models in the third and fourth steps, including the initial model and the adjusted models, respectively. The final parsimonious model adjusted for confounders that were significantly associated with the dependent variables.

### 2.4. Ethical Statements

This study was conducted in accordance with the Declaration of Helsinki, and the study protocol was reviewed and approved by the Institutional Review Board of School of Health Science and Faculty of Medical Sciences, Wuhan University (IRB number: 2019YF2050). Informed consent information was included with each questionnaire and introduced before the surveys.

Surveys were only conducted if subjects were fully informed of the content and aim of this research project and agreed to participate. The survey was also conducted anonymously, and respondents’ information was kept confidential and only for the use of scientific research. 

## 3. Results

### 3.1. Descriptions of Sample Characteristics

A total of 4993 elderly people was assessed in the study, while there were some missing values. As shown in Table 1, 83.30% of the sample participants were between 60 and 79 years, with a lower proportion of males (49.70%) and individuals with an BMI <18.50 or BMI ≥24.00 (37.09%). Of those eligible participants, more than half received less than five years of education, 69.61% came from rural areas, and 63.87% were married or cohabiting. In terms of family financial situation and health status, around one-third of the subjects self-reported on these characteristics as poor, which accounted for 28.49% and 39.62%, respectively. 

### 3.2. Association between the Number of Health-Related Risk Behaviors and Sleep Disturbances

Among the participants, 2285 (45.76%) had sleep disturbances. The results of the chi-square tests and logistic regression analyses examining the cross-sectional relationship between the number of health-related risk behaviors and sleep disturbances are presented in Table 2 and Table 3. In Table 2, the Pearson chi-square test showed that there was a difference between sleep disturbances and health-risk behaviors (χ^2^ = 203.703, *p* < 0.001), and the linear trend chi-square test indicated the prevalence of sleep disturbances was higher with an increase of health-risk behaviors (χ^2^ = 139.969, *p* < 0.001). To further explore their cross-sectional relationship, logistic regression analyses were conducted (Table 3). As compared with the elderly without any health-risk behaviors, the odds ratios (ORs) of sleep disturbances for those with one to five health-risk behaviors were 1.72 (95% CI = 1.39 to 2.13), 2.81 (95% CI = 2.28 to 3.47), 3.76 (95% CI = 3.02 to 4.68), 2.57 (95% CI = 2.02 to 3.26), and 4.02 (95% CI = 2.88 to 5.60), respectively, in a crude model. After adjusting for the predictors that were significantly associated with sleep disturbances (model 3), more health-related risk behaviors were still significantly associated with a higher prevalence of sleep disturbances.

### 3.3. Association between the Number of Health-Related Risk Behaviors and Sleep Quality

According to the result of the t test, individuals with sleep disorders scored lower in self-perceived sleep quality (*t* = 48.684, *p* < 0.001). Multiple linear regression analyses were conducted to determine if the number of health-related risk behaviors were associated with self-perceived sleep quality (Table 4). In crude and adjusted models, more health-related risk behaviors were significantly and constantly related to worse self-perceived sleep quality. In particular, as compared with the reference group (with zero health-risk behaviors), individuals with one health-related risk behavior (B = 0.14, 95% CI = −0.23 to −0.06, *p* < 0.01), two health-related risk behaviors (B = 0.21, 95% CI = −0.30 to −0.13, *p* < 0.001), three health-related risk behaviors (B = 0.46, 95% CI = −0.55 to −0.37, *p* < 0.001), four health-related risk behaviors (B = 0.50, 95% CI = −0.62 to −0.39, *p* < 0.001), and five health-related risk behaviors (B = 0.83, 95% CI = −1.00 to −0.66, *p* < 0.001) showed lower scores of self-perceived sleep quality in model 3, after adjusting for the confounding factors that were significantly associated with sleep quality.

## 4. Discussion

As the most populated country, China has the largest elderly population in the world. However, there are only a few sleep-related studies specifically targeted at the elderly in China. In this study, 45.76% of the respondents self-reported having at least one sleep problem. The percentage is obviously high, which may be attributed to the method used in this study to measure sleep problems. In addition, findings of a comprehensive meta-analysis reported by researchers at the University of Macau [28] showed that the prevalence of sleep disturbances among Chinese older adults in rural areas was 44.0%, which is very similar to the result of this study. Furthermore, the fact is that most of the participants in this study were also from rural areas, but this is in line with the actual situation in China. It is easy to understand that elderly individuals who self-reported having at least one type of sleep problem self-reported significantly worse sleep quality than older adults who do not have any sleep problems.

Older people are considered to have a higher prevalence of sleep-related, health-related risk issues [28,29,31]. Previous studies from developed countries or regions have confirmed the link between specific behaviors and sleep health. Health-related behavioral risk factors, including tobacco use, alcohol drinking, unhealthy dietary behaviors, as well as insufficient time for leisure activities and physical inactivity, have been shown to be associated with sleep health [17,19,20,32,33]. The results of these studies provide the theoretical basis for developing interventions that specifically target a particular health-related risk behavior. From the perspective of health-related behavioral interventions, the effect of one single health-related risk behavior intervention might fail to live up to its expectation, since the health-related risk behaviors are interrelated with each other in most cases. Alternative approaches for developing the appropriate joint intervention strategies that combine multiple health-related risk behaviors could be utilized on the basis of determining the relationship between the number of daily health-related behavioral risk factors and sleep among the elderly.

As expected, after adjustment for predictors that were significantly associated with the participants’ sleep disturbances, results of the final parsimonious model noted that those Chinese older adults with one to five health-related risk behaviors had 1.41, 2.09, 2.54, 2.12, and 2.49 odds of suffering from sleep disturbances. In general, the test for linear trend suggested that the adjusted odds ratios increased with the number of health-related risk behaviors. Therefore, it is reasonable to assume that the risk of sleep disturbances among Chinese older adults increases with the number of daily health-related behavioral risk factors. In addition, the results of this study showed that the association between the number of health-related risk behaviors and level of self-perceived sleep quality was strongly positively correlated among Chinese older adults. The linear trend test revealed that individuals with a higher number of health-related risk behaviors were more likely to self-report poorer sleep quality.

Substantial concrete evidence indicated that there was a significant negative effect on the individual’s sleep health when people engaged in health-related risk behavior(s) [19,20,32,33]. This study concluded that with an increase in health-related risk behaviors, the negative effects on sleep health could be further increased, at least for both the increased prevalence of sleep problems and the deterioration of sleep quality. Currently, it is known that most existing behavioral intervention programs are still focused on the prevention and control of one single health-related risk behavior. The effects of an isolated behavior-specific modification program may be counteracted by interconnections with other health-related risk behaviors because health-related risk behaviors usually do not occur at random but often manifest themselves in specific patterns of combinations [34,35].

Finally, the results of this study indicated the possibility that behavioral interventions and programs could be helpful for improving the quality of sleep and reducing the prevalence of sleep problems or sleep disorders for the elderly. Meanwhile, and more importantly, such interventions and programs should not ignore the importance of how to prevent or reduce older adults from engaging in multiple health-related risk behaviors. Our findings suggest that joint interventions and programs that aim to scientifically and effectively prevent and control the co-occurrence of multiple health-related risk behaviors among the older people may be urgently needed. A general package, compiled and combined with multiple related intervention programs across the full range of health-related risk behaviors, should be designed and implemented immediately to generate substantially larger health benefits as a favorable and cost effective alternative.

## 5. Limitations

Although this is the first study to address the cross-sectional relationship between daily health-related behavioral risk factors and sleep among the elderly, in China, our findings need to be considered in light of some limitations. First, variables, including daily health-related behaviors, prevalence of sleep problems, and sleep quality were self-reported, although the validity of these measures has been well-established with respect to objective measures of epidemiological studies and subjective reports of health outcomes, for example, sleep problems were commonly used in conventional epidemiological studies [29,32,36]. Secondly, recall bias due to false or inaccurate responses from the participants could have played a role in our results. Third, causal effects cannot be ascertained due to the cross-sectional design. Fourth, China’s Health-Related Quality of Life Survey for Older Adults 2018 used standardized questions to measure sleep problems in the past 30 days without reporting the frequency. Hence, future studies are essential in order to confirm the findings from this study.

## 6. Conclusions

This study has clearly shown that having multiple health-related risk behaviors was positively correlated with a higher risk of sleep disturbances among Chinese elderly. Moreover, elderly individuals with multiple health-related risk behaviors were significantly associated with lower scores of self-perceived sleep quality. Our findings indicate the possibility that behavioral interventions and programs could be helpful for improving the quality of sleep and reducing the prevalence of sleep problems or sleep disorders for the elderly. Meanwhile, and more importantly, such interventions and programs should not ignore the importance of preventing or reducing the number of older adults that engage in multiple health-related risk behaviors. Joint interventions and programs that aim to scientifically and effectively prevent and control the co-occurrence of multiple health-related risk behaviors among the older people are urgently needed.

## Figures and Tables

**Table 1 ijerph-16-04905-t001:** Descriptions of the general demographic characteristics of the study population.

Characteristics	Categories	Frequency (*n*)	Percentage (%)
Age (years)	60–64	1113	22.35
	65–69	1171	23.51
	70–74	1143	22.95
	75–79	721	14.48
	≥80	832	16.70
Sex	Male	2465	49.70
	Female	2495	50.30
Body mass index (BMI)	<18.50	507	10.41
	18.50–23.99	3063	62.91
	≥24.00	1299	26.68
Years of education	0	1105	23.81
	1–5	1397	30.10
	6–8	940	20.25
	9–11	650	14.01
	>11	549	11.83
Household registration	Countryside	3422	69.61
	City	1494	30.39
Marital status	Married/cohabiting	3169	63.87
	others	1793	36.13
Self-rated family financial situation	Poor	1417	28.49
	Fair	2955	59.42
	Rich	601	12.09
Self-rated health status	Poor	1976	39.62
	General	2341	46.93
	Good	671	13.45

Note: Sample sizes of the demographic characteristic variables may not sum to *n* = 4993 due to missing values.

**Table 2 ijerph-16-04905-t002:** Distribution of the number of health-related risk behaviors among participants with and without sleep disturbances.

Number of Health-Related Risk Behaviors	No (*n* = 2708, 54.24%)	Yes (*n* = 2285, 45.76%)	χ^2^/ *t*
0	450 (73.77)	160 (26.23)	203.703 a*** 139.969 b***
1	754 (62.06)	461 (37.94)	
2	648 (50.00)	648 (50.00)	
3	451 (42.79)	603 (57.21)	
4	321 (52.28)	293 (47.72)	
5	84 (41.18)	120 (58.82)	
Sleep quality (mean ± SD)	3.86 ± 0.78	2.73 ± 0.81	48.684 ***

Notes: a, Pearson chi-square; b, linear trend chi-square; *** *p*-value < 0.001.

**Table 3 ijerph-16-04905-t003:** Odds ratio for the number of health-related risk behaviors among participants with and without sleep disturbances.

Number of	Model 1, OR (95% CI)	Model 2, OR (95% CI)	Model 3, OR (95% CI)
0	1.00 (Reference)	1.00 (Reference)	1.00 (Reference)
1	1.72 (1.39,2.13) ***	1.40 (1.10–1.78) **	1.41 (1.11–1.78) **
2	2.81 (2.28–3.47) ***	2.03 (1.60–2.58) ***	2.09 (1.66–2.63) ***
3	3.76 (3.02–4.68) ***	2.53 (1.96–3.26) ***	2.54 (1.99–3.25) ***
4	2.57 (2.02–3.26) ***	1.92 (1.44–2.57) ***	2.12 (1.60–2.80) ***
5	4.02 (2.88–5.60) ***	2.31 (1.56–3.43) ***	2.49 (1.70–3.65) ***

Notes: * *p*-value < 0.05, ** *p*-value < 0.01, and *** *p*-value < 0.001. Model 1 is the crude model. Model 2 adjusted for age, gender, BMI, years of education, household registration, marital status, self-reported economic status, and self-reported health status. Model 3 is the final parsimonious model, adjusted for predictors that were significantly associated with the participants’ sleep disturbances (sex, marital status, years of education, self-rated family financial situation, and self-rated health status).

**Table 4 ijerph-16-04905-t004:** Coefficients (unstandardized) for the number of health-related risk behaviors among participants with different levels of self-perceived sleep quality.

Number of	Model 1, B (95% CI)	Model 2, B (95% CI)	Model 3, B (95% CI)
0	1.00 (reference)	1.00 (reference)	1.00 (reference)
1	−0.26 (−0.35, −0.17) ***	−0.14 (−0.23, −0.06) **	−0.14 (−0.23, −0.06) **
2	−0.42 (−0.51, −0.33) ***	−0.22 (−0.30, −0.13) ***	−0.21 (−0.30, −0.13) ***
3	−0.73 (−0.82, −0.64) ***	−0.47 (−0.56, −0.38) ***	−0.46 (−0.55, −0.37) ***
4	−0.71 (−0.82, −0.59) ***	−0.51 (−0.62, −0.39) ***	−0.50 (−0.62, −0.39) ***
5	−1.12 (−1.29, −0.96) ***	−0.86 (−1.03, −0.69) ***	−0.83 (−1.00, −0.66) ***

Notes: * *p*-value < 0.05; ** *p*-value < 0.01; and *** *p*-value < 0.001. Model 1 is the crude model. Model 2 adjusted for age, gender, BMI, years of education, household registration, marital status, self-reported economic status, and self-reported health status. Model 3 is the final parsimonious model, adjusted for predictors that were significantly associated with the participants’ self-perceived sleep quality (age, sex, years of education, household registration, marital status, self-rated family financial situation, and self-rated health status).

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
