# Peer review of "The Relationship between the Number of Daily Health-Related Behavioral Risk Factors and Sleep Health of the Elderly in China"

_ijerph, 2019, doi:10.3390/ijerph16244905_

Round 1

Reviewer 1 Report

Relationship between the number of daily health-related behavioral risk factors and sleep health among the elderly in China

General comments: I agree with the authors that their research question is an important one, especially for an ageing population. However, the way sleep terminology has been used here must be changed and be consistent across the entire paper. The term ‘sleep disorders’ has been used inappropriately and I strongly recommend either removing that form this article, as it is unclear how it was measured and I doubt its validity.

Abstract

If word count allows specify how sleep disorder was measured. Please say this was a cross-sectional analysis.

Introduction

2 line 40-42: “Good sleep quality can not only eliminates tiredness and improve memory, but also regulates body functions and therefore 41 maintain health. However, sleep disorders are extremely common…” please, in the entire paper, be careful in how you use sleep terminology. Sleep problems and too short/too long sleep are common but this is not the same about sleep disorders such as insomnia (see Morin, C. M., LeBlanc, M., Daley, M., Gregoire, J. P., & Merette, C. (2006). Epidemiology of insomnia: prevalence, self-help treatments, consultations, and determinants of help-seeking behaviors. Sleep medicine, 7(2), 123-130.). So please do not use the term ‘sleep disorders’ unless you are referring to insomnia disorder or sleep apnoea.

On page 2 you twice say it was ‘proved’. Can you turn it down and say, for example, shown or suggested or even reported? This comment refers also to the entire article.

Method

How was insufficient intake of vegetable and fruit defined?

Can you clarify if in the question “The respondents were asked to answer the question as “whether you had the following sleep problems in the past 30 days: 1) do not have any sleep problem; 2) have trouble falling asleep; 3) dreaminess; 4) frequently wake up at night; 5) oversleep; 6) others” participants could specify how often in the past 30 days they experienced each sleep problem? Otherwise this would lead to extremely high prevalence of sleep problems as participants with e.g. one sleep complaint will be in the same group as someone with daily sleep difficulties.

Results

Page 5, line 176 “Among the participants, 2,285 (45.76%) had sleep disorders.” If you look at the literature on prevalence of sleep apnoea (e.g. Punjabi, N. M. (2008). The epidemiology of adult obstructive sleep apnea. Proceedings of the American Thoracic Society, 5(2), 136-143 ) and insomnia (Morin, C. M., LeBlanc, M., Daley, M., Gregoire, J. P., & Merette, C. (2006). Epidemiology of insomnia: prevalence, self-help treatments, consultations, and determinants of help-seeking behaviors. Sleep medicine, 7(2), 123-130.), two most common sleep disorders, it is not possible for the prevalence of sleep disorders to be this high. Please provide more justification, or remove this from the paper. Crucially, the authors never actually specify which disorders were actually reported in this study.

Discussion

On page 5 the authors wrote “Results of a comprehensive meta-analysis reported by researchers at the University of Macau [28] showed that the prevalence of sleep disturbances among Chinese older adults in rural areas was 44.0%, which is very similar to the result of this study.” I assume the authors refer here to their ‘sleep disorders prevalence’? I strongly recommend they use the term sleep disturbances rather than disorders as this is an inaccurate term for what was measured here.

P 8 line 257 “ concrete evidences “ evidence not evidences

P 8 Lines 236 -260 please avoid repeating yourself

In your limitations you say that “Thirdly, most of our participants are from rural areas, but this is in line with the actual situation in China” . Why is this a limitation? In fact I would like to see this discussed in the paper, and compared with other studies.

You need to acknowledged this was a cross-sectional study.

Author Response

Responses to the reviewers’ comments
Reviewer #1
COMMENTS
General comments: I agree with the authors that their research question is an important
one, especially for an ageing population. However, the way sleep terminology has been
used here must be changed and be consistent across the entire paper. The term ‘sleep
disorders’ has been used inappropriately and I strongly recommend either removing that
form this article, as it is unclear how it was measured and I doubt its validity.
Response: We thank and agree with the reviewer for this professional comment. We have
removed the term “sleep disorders” form this article, and we use the term “sleep
problems” consistently throughout the article.
Comment #1) Abstract
If word count allows specify how sleep disorder was measured. Please say this was a
cross-sectional analysis.
Response: We have made this revision.
Comment #2) Introduction
2 line 40-42: “Good sleep quality can not only eliminates tiredness and improve memory,
but also regulates body functions and therefore 41 maintain health. However, sleep
disorders are extremely common…” please, in the entire paper, be careful in how you use
sleep terminology. Sleep problems and too short/too long sleep are common but this is
not the same about sleep disorders such as insomnia (see Morin, C. M., LeBlanc, M.,
Daley, M., Gregoire, J. P., & Merette, C. (2006). Epidemiology of insomnia: prevalence,
self-help treatments, consultations, and determinants of help-seeking behaviors. Sleep
medicine, 7(2), 123-130.). So please do not use the term ‘sleep disorders’ unless you are
referring to insomnia disorder or sleep apnoea.
Response: We thank the reviewer for this excellent suggestion, and we've made the
changes accordingly.
Comment #3) On page 2 you twice say it was ‘proved’. Can you turn it down and say, for
example, shown or suggested or even reported? This comment refers also to the entire
article.
Response: We have made this edit.
Comment #4) How was insufficient intake of vegetable and fruit defined?
Response: The China's Health-Related Quality of Life Survey for Older Adults 2018
didn't ask about the specific intake of vegetables and fruits. Participants were asked to
self-report whether they skip breakfast and/ or have an unbalanced diet like insufficient
intake of vegetable and fruit. This is a limitation of this study, which we have mentioned
in the Limitation section.
Comment 5) Can you clarify if in the question “The respondents were asked to answer the
question as “whether you had the following sleep problems in the past 30 days: 1) do not
have any sleep problem; 2) have trouble falling asleep; 3) dreaminess; 4) frequently wake
up at night; 5) oversleep; 6) others” participants could specify how often in the past 30
days they experienced each sleep problem? Otherwise this would lead to extremely high
prevalence of sleep problems as participants with e.g. one sleep complaint will be in the
same group as someone with daily sleep difficulties.
Response: We appreciate this thoughtful comment. But the questionnaire used
standardized questions “yes-no” answers without reporting their frequency. This is
another limitation, which we have added in the discussion to explain the results, and also
mentioned in the Limitation section.
Comment #6) Results
Page 5, line 176 “Among the participants, 2,285 (45.76%) had sleep disorders.” If you
look at the literature on prevalence of sleep apnoea (e.g. Punjabi, N. M. (2008). The
epidemiology of adult obstructive sleep apnea. Proceedings of the American Thoracic
Society, 5(2), 136-143 ) and insomnia (Morin, C. M., LeBlanc, M., Daley, M., Gregoire, J.
P., & Merette, C. (2006). Epidemiology of insomnia: prevalence, self-help treatments,
consultations, and determinants of help-seeking behaviors. Sleep medicine, 7(2),
123-130.), two most common sleep disorders, it is not possible for the prevalence of sleep
disorders to be this high. Please provide more justification, or remove this from the paper.
Crucially, the authors never actually specify which disorders were actually reported in
this study.
Response: A similar comment was introduced in Comment 5. We have removed the term
“sleep disorders” form this article, and we use the term “sleep problems” consistently
throughout the article.
Comment #7) Discussion
On page 5 the authors wrote “Results of a comprehensive meta-analysis reported by
researchers at the University of Macau [28] showed that the prevalence of sleep
disturbances among Chinese older adults in rural areas was 44.0%, which is very similar
to the result of this study.” I assume the authors refer here to their ‘sleep disorders
prevalence’? I strongly recommend they use the term sleep disturbances rather than
disorders as this is an inaccurate term for what was measured here.
Response: We thank the reviewer for bringing up this advice. We have made the change,
as the reviewer suggests.
Comment 8) P 8 line 257 “ concrete evidences “ evidence not evidences.
Response: We have made this edit.
Comment 9) P 8 Lines 236 -260 please avoid repeating yourself.
Response: We have made the change accordingly.
Comment 10) In your limitations you say that “Thirdly, most of our participants are from
rural areas, but this is in line with the actual situation in China” . Why is this a limitation?
In fact I would like to see this discussed in the paper, and compared with other studies.
Response: We have made this revision.
Comment 11) You need to acknowledged this was a cross-sectional study.
Response: We have made this edit to make it clearer now.
Again, we thank the reviewer for all these valuable review comments.

Reviewer 2 Report

It is an important article about health behavior and sleep quality in China, the research question is nicely introduced, methods and results are carefully presented and discussed. There are only minor remarks.

Introduction: carefully done 

Method: response rate is missing

Results: 

1) In line 169 the word abnormal should be omitted. Clarify; how many have overweight and underweight

2) Descriptive information about the outcome (sleep quality) is missing. Please add in table 1.

3) Please add more specific information on health risk behaviour. It would be nice to get descriptive information about the frequency of each single risky behaviour. When this table is added to the article, it might be possible to suggest which risky behaviour most importantly contributes to sleep quality:

Author Response

Reviewer #2
COMMENTS
General comments: It is an important article about health behavior and sleep quality in
China, the research question is nicely introduced, methods and results are carefully
presented and discussed. There are only minor remarks.
Response: We thank the reviewer for the supportive and valuable review comment.
Comment #1) Introduction: carefully done.
Response: We appreciate this comment.
Comment #2) Method: response rate is missing.
Response: We have made this edit.
Comment #3) Results:
In line 169 the word abnormal should be omitted. Clarify; how many have overweight
and underweight.
Response: We have made this revision.
Comment #4) Descriptive information about the outcome (sleep quality) is missing.
Please add in table 1.
Response: Descriptive information about the sleep quality are presented in Tables 2.
Comment #5) Please add more specific information on health risk behaviour. It would be
nice to get descriptive information about the frequency of each single risky behaviour.
When this table is added to the article, it might be possible to suggest which risky
behaviour most importantly contributes to sleep quality.
Response: We appreciate this professional advice. We suspect that it could be differences
from different research perspectives and aims. Since the present study aimed to examine
the cross-sectional association between sleep health and the number of health-related
behaviors.
Nonetheless, this study does not intend to determine which risky behaviour or patterns
maybe the most importantly contributes to sleep health. We, therefore, did not show the
results of the information about the frequency of each single risky behaviour in the
present form. (They’ll be presented in another article by latent class analysis.)
Again, we very much appreciate the thoughtful and critical feedback from the reviewer

Round 2

Reviewer 1 Report

Please fully adress comments 4, 5 and 10.  I cannot see them being mentioned in the limitation section, or explained in the Results.  The way you measure (and explain) sleep is fundamental to the scientific credence of this paper. 

Author Response

Responses to the reviewers’ comments
COMMENTS
Please fully adress comments 4, 5 and 10. I cannot see them being mentioned in the
limitation section, or explained in the Results. The way you measure (and explain) sleep
is fundamental to the scientific credence of this paper.
Comment 4) How was insufficient intake of vegetable and fruit defined?
Response: The China's Health-Related Quality of Life Survey for Older Adults 2018
didn't ask about the specific intake of vegetables and fruits. Participants were asked to
self-report whether they skip breakfast and/ or have an unbalanced diet like insufficient
intake of vegetable and fruit. This is a limitation of this study, which we have mentioned
in the Limitation section.
The first question is about skipping breakfast, The question is “How often do you skip
breakfast? (1) Never, I have it every day; (2) I rarely skip breakfast; (3) I occasionally
skip breakfast; (4) I often skip breakfast (5) Basically, I almost skip breakfast every day”.
Another question about eating habits is “Generally speaking, is your diet balanced? (1)
Yes, I basically have a balanced diet of fruits, vegetables and meat; (2) No, I mainly eat
meat and rarely eat fruits and/ or vegetables; (3) No, I mainly eat fruits and vegetables,
basically do not eat meat”.
In this study, participants answered he/ she (3) occasionally skip breakfast/ (4) often skip
breakfast/ (5) almost skip breakfast every day” and/ or “(2) mainly eat meat and rarely eat
fruits and/ or vegetables/ (3) mainly eat fruits and vegetables, basically do not eat meat”
was identified as unhealthy dietary behaviors.
Please see the revised text in the Methods section “2.2.2. Assessment of health risk
behaviors”: 3) unhealthy dietary behaviors: individuals who self-reported to skip
breakfast or have an unbalanced diet like insufficient intake of vegetable and fruit.
As for the bias that such subjective reporting may lead to, we have written in the
limitations section that “Firstly, variables, including daily health-related behaviors,
prevalence of sleep problems, and sleep quality were self-reported, even though the
validity of these measures have been well established with respect to objective measures
of epidemiological studies, and subjective reports of health outcomes, like sleep problems
were commonly used in conventional epidemiological studies. [29, 32, 36] Secondly,
recall bias due to false or inaccurate responses from the participants may play a role in
our results.”
Comment 5) Can you clarify if in the question “The respondents were asked to answer the
question as “whether you had the following sleep problems in the past 30 days: 1) do not
have any sleep problem; 2) have trouble falling asleep; 3) dreaminess; 4) frequently wake
up at night; 5) oversleep; 6) others” participants could specify how often in the past 30
days they experienced each sleep problem? Otherwise this would lead to extremely high
prevalence of sleep problems as participants with e.g. one sleep complaint will be in the
same group as someone with daily sleep difficulties.
Response: We appreciate this thoughtful comment. But the questionnaire used
standardized questions “yes-no” answers without reporting their frequency.
Here's the English translation of the question.
Do you had the following sleep problems in the past 30 days Yes No
1 I do not have any sleep problem
2 have trouble falling asleep
3 dreaminess
4 frequently wake up at night
5 oversleep
6 Others, specify: ___________
This is another limitation, which we have added in the discussion to explain the results,
and also mentioned in the Limitation section.
We have added more detail to the Discussion section: “In the present study, 45.76% of
the respondents self-reported having at least one sleep problem. The percentage is
obviously high, which may be attributed to the method used in this study to measure sleep
problems. Yet, findings of a comprehensive meta-analysis reported by researchers at the
University of Macau [28] showed that the prevalence of sleep disturbances among
Chinese older adults in rural areas was 44.0%, which is very similar to the result of this
study. Furthermore, the fact is that most of the participants in this study were also from
rural areas, but this is in line with the actual situation in China.”
We also mentioned in the Limitation section: “Firstly, variables, including daily
health-related behaviors, prevalence of sleep problems, and sleep quality were
self-reported, even though the validity of these measures have been well established with
respect to objective measures of epidemiological studies, and subjective reports of health
outcomes, like sleep problems were commonly used in conventional epidemiological
studies. [29, 32, 36]”
and “Fourthly, the China's Health-Related Quality of Life Survey for Older Adults 2018
used standardized questions to measure sleep problems in the past 30 days without
reporting their frequency. ”
Comment 10) In your limitations you say that “Thirdly, most of our participants are from
rural areas, but this is in line with the actual situation in China” . Why is this a limitation?
In fact I would like to see this discussed in the paper, and compared with other studies.
Response: We've removed these statements from the Limitation section.
We wrote in the Discussion section: “Furthermore, the fact is that most of the participants
in this study were also from rural areas, but this is in line with the actual situation in
China.” What’s more, sleep related health issue is overlooked in China. No national data
on the prevalence of sleep health, like insomnia, sleep apnoea, and other sleep disorders
have been reported, nor have studies specifically targeted at sleep problems of the elderly
been conducted.
Again, we very much appreciate the thoughtful and critical feedback from the reviewer.

Round 3

Reviewer 1 Report

Thank you for addressing my comments.